# Effects of Neural Network Parameter Pruning on Uncertainty Estimation

## Abstract

Obtaining uncertainty estimates for deep neural network predictions is a particularly interesting problem for safety-critical applications. In addition, uncertainty estimation for compressed neural networks is a problem that is well-suited to real-world applications. At the same time, the combination of uncertainty estimation with parameter pruning is far from being well understood. In this work, we present a study on the influence of parameter pruning on the uncertainty estimation of a deep neural network for image classification. We compress two popular image classification networks with five pruning approaches and investigate the uncertainty estimation next to the standard accuracy metric after pruning. To measure the uncertainty performance, we propose a new version of the Area Under the Sparsification Error (AUSE) for image classification and additionally evaluate using the excepted calibration error (ECE). We study the influence of pruning on the standard uncertainty estimation methods maximum softmax probability, Monte Carlo dropout, and deep ensembles using two datasets, namely CIFAR-100 and ImageNet. The analysis shows that pruning affects, besides the class prediction, also the uncertainty estimation performance of a neural network in a negative manner. In general, the uncertainty estimation performance decreases with an increasing pruning sparsity, similar to the class prediction. Noticeably, in some special cases, the pruning can improve the neural network's uncertainty estimation. Our code will be published after acceptance.

## 1 Introduction

Deep neural networks have made significant progress in computer vision (He et al., 2016; Rudolph et al., 2022; Simonyan & Zisserman, 2015; Bouazizi et al., 2022). Additionally, their execution time remains low, thanks to model compression, making them suitable for real-world applications. However, the deployment of such algorithms in safety-critical applications, such as automated driving (Hawke et al., 2020; Wiederer et al., 2020), robotics (Jang et al., 2022), and medical image analysis (Dawoud et al., 2021), raises questions regarding the lack of reliability measures for the predictions. In particular, uncertainty measures enable the removal of erroneous model predictions, but most current state-of-the-art approaches only provide the prediction. Importantly, the impact of model compression on uncertainty estimates has rarely been explored in the past.

Two common methods for compressing neural networks are parameter pruning and parameter quantization. Parameter pruning, reduces the number of the neural network parameters by eliminating redundant and insignificant ones, results in decreased hardware requirements (Li et al., 2017; Liu et al., 2017; Lin et al., 2020; Ding et al., 2021). Parameter quantization decreases the bit-width of the model's weights to enable efficient inference (Défossez et al., 2022). Although previous research (Ferianc et al., 2021) has addressed the impact of quantization on uncertainty estimation, no study has yet examined this issue in the context of parameter pruning. While, the pruning effect on the model's predictive performance is well-studied in the community (Li et al., 2017; Lin et al., 2020; Ding et al., 2021) and even the increased robustness of the pruned neural network is proven in literature (Li et al., 2023), the effect of pruning on the uncertainty estimates remains unexplored.

Our work investigates the effect of parameter pruning on uncertainty estimation. We consider the task of image recognition with deep neural networks that additionally provide uncertainty estimates. Several parameter pruning approaches are combined with different uncertainty estimation techniques. We then evaluate not only the predictive class output, but also the uncertainty measures. We perform our extensive evaluation on two image classification benchmarks, namely the CIFAR-100 (Krizhevsky et al., 2009) and the large-scale dataset ImageNet (Deng et al., 2009), using different standard neural network architectures. In particular, the models are pruned using random pruning, L1 pruning (Li et al., 2017), Batch Normalization pruning (Liu et al., 2017), HRank pruning (Lin et al., 2020), and ResRep pruning (Ding et al., 2021). In addition, we explore the impact of pruning on the uncertainty performance of maximum softmax probability (MSP) (Hendrycks & Gimpel, 2017), bootstrapped ensembles (Lakshminarayanan et al., 2017), and Monte Carlo (MC) dropout (Gal & Ghahramani, 2015). We introduce the Area Under the Sparsification Error (AUSE) metric, based on optical flow metric (Ilg et al., 2018) to quantify the uncertainty performance. Next to the proposed evaluation with the AUSE metric, the excepted calibration error (ECE) (Naeini et al., 2015) is used as a standard metric to evaluate classification problems. The experiments prove that pruning a neural network not only affects its class prediction but also negatively impacts the uncertainty estimation. In general, with an increasing pruning sparsity, both the uncertainty estimation and class prediction performance of the neural network decrease. However, there are some cases where pruning can improve the neural network's uncertainty estimation performance.

Our study makes three main contributions. First, we investigate the impact of pruning on uncertainty prediction in deep neural networks, a problem that has not yet been explored. To this end, we provide a comprehensive analysis of *two* common neural network architectures, *five* pruning techniques, and *three* uncertainty estimation methods on *two* image classification datasets. Secondly, we present the Area Under the Sparsification Error (AUSE) metric as an uncertainty measure for image classification to quantify uncertainty prediction. We evaluate the capability of the pruned models to predict uncertainties using the introduced metric. Thirdly, the exploration of the excepted calibration error (ECE) and the AUSE indicates that pruning may enhance the calibration and uncertainty estimation of the neural network, depending on the architecture and uncertainty estimation approach. In hindsight, we formulate our claim that model pruning affects the model performance along with the uncertainty estimation. We validate our claims with our experimental analysis, provided in Section 6 of this paper.

## 2 Related Work

### 2.1 Parameter Pruning

Parameter pruning removes unimportant or redundant parameters of a neural network to lower resource consumption during inference. A distinction is made between unstructured pruning methods, which remove individual weights (Han et al., 2015) or groups of weights (Wen et al., 2016) and structured pruning methods that discard entire network channels (Liu et al., 2017; Lin et al., 2020; Liu et al., 2021). We focus on structured pruning because unstructured pruning leads to sparse weights, and the execution of sparse weights requires special hardware and software. Simple pruning approaches rely only on the network's internal knowledge and use the L1-norm of the model parameters (Li et al., 2017) or the Batch Normalization (Ioffe & Szegedy, 2015) parameters (Liu et al., 2017; 2021) to measure the channel's importance. Liu et al. (Liu et al., 2017) determine the channels to prune with the Batch Normalization's scaling weights, and GFBS (Liu et al., 2021) expands this by using the Batch Normalization weights together with a Taylor expansion. Instead of the Taylor expansion, You et al. (You et al., 2022) use a spline interpretation of the neural network to define the parameters to prune. At the same time, other methods rely not only on internal knowledge but also consider the input data's effect (Lin et al., 2020; Duan et al., 2022). Lin et al. (Lin et al., 2020) use the matrix rank of the activation's output feature maps to determine the channel's importance. Due to the parameter reduction, some knowledge gets lost during pruning which can be recovered with fine-tuning. In some approaches, all parameters are pruned at once, and at the end, one fine-tuning step is performed (Li et al., 2017; Liu et al., 2017), or the network is pruned layer-wise with intermediate fine-tuning steps (Lin et al., 2020). To prevent knowledge reduction, ResRep (Ding et al., 2021) categorizes the channels during fine-tuning with the help of additional layers in remembering and forgetting parts and removes the forgetting parts without

a performance drop. While the former methods all require training data for fine-tuning, DFNP (Tang et al., 2021) and DFBF (Holzbock et al., 2023) work with synthetic training data, and iSparse (Garg & Candan, 2020) prunes without the necessity of fine-tuning. The application of most pruning approaches is limited to a specific task and architecture. To solve this, DFBF (Holzbock et al., 2023) proposes a method applicable to various computer vision tasks, and DepGraph (Fang et al., 2023) is able to prune different neural network architectures whitout an adaption of the pruning algorithm. Existing methods only consider the prediction performance of the pruned models. Nevertheless, the influence of pruning on uncertainty estimates has not been considered. Therefore, the focus of our work is the performance of various pruning approaches on uncertainty estimation.

## 2.2 Uncertainty Estimation

Uncertainty in deep neural networks arises from uncertainty inherent in the data (aleatoric) or uncertainty due to a lack of knowledge, which is reducible with additional training data (epistemic) (Kendall & Gal, 2017). Empirical approaches (Lakshminarayanan et al., 2017; Gal & Ghahramani, 2015; Huang et al., 2017) place a distribution over the model weights and use the resulting output mean and variance to estimate the epistemic uncertainty. Bootstrapped ensembles (Lakshminarayanan et al., 2017) approximate the weight distribution by training multiple models with different initialization, whereas Snapshot ensembles (Huang et al., 2017) leverage cyclic learning rate scheduling to reduce the training effort. On the other hand, MC dropout (Gal & Ghahramani, 2015) applies dropout (Srivastava et al., 2014) during inference at different network layers to approximate the weight distribution. Instead of sampling the epistemic uncertainty, DEUP (Lahlou et al., 2023) defines it based on the aleatoric uncertainty and the generalization error. Another method for epistemic uncertainty estimation uses the reconstruction error between the original input image and the reconstructed image (Hornauer et al., 2023). Predictive approaches (Hendrycks & Gimpel, 2017; Nix & Weigend, 1994) assume a prior distribution over the model output. The built-in MSP (Hendrycks & Gimpel, 2017) is a simple predictive approach that targets the aleatoric uncertainty in image classification. Other predictive methods, in contrast, assume a Gaussian (Nix & Weigend, 1994) or evidential (Amini et al., 2020) output distribution that is used as a loss function for optimization, which is not applicable to classification. Mukhoti et al. (Mukhoti et al., 2021) address both types of uncertainty in one framework by combining the softmax probability with the Gaussian Mixture Model of the feature density. Other works target uncertainty estimation for high-dimensional tasks like monocular depth estimation (Poggi et al., 2020; Hornauer & Belagiannis, 2022), optical flow estimation (Ilg et al., 2018), semantic segmentation (Kendall et al., 2017; Kendall & Gal, 2017), or trajectory prediction (Wiederer et al., 2023). While several uncertainty estimation approaches have been proposed in recent years and the impact of quantization on uncertainty estimation has been evaluated (Ferianc et al., 2021), the influence of pruning on uncertainty quantification has not yet been studied. The crucial difference is that quantization lowers the computational effort by reducing the bit-width, and pruning removes entire parameters.

# 3 Background Concepts

This section gives an overview of the applied pruning and uncertainty methods, which are later combined to investigate the effect of pruning on uncertainty estimation.

## 3.1 Parameter Pruning

Pruning aims to reduce neural network parameters while retaining the model's performance. We use five pruning approaches to investigate the effect of pruning algorithms on the uncertainty estimation of a neural network. In the following, we give an overview of the utilized pruning algorithms, namely Random Pruning, L1 Pruning (Li et al., 2017), Batch Normalization Pruning (Liu et al., 2017), HRank Pruning (Lin et al., 2020), and ResRep Pruning (Ding et al., 2021).

**Pruning Definition**    Given the trained deep neural network $f(\cdot)$, with parameters $\theta$ and $N$ different layers, we focus on channel pruning, where output channels are removed from the network layers. To prune a single convolution, we consider the convolutional layer $\mathbf{z}_n = Conv_n(\mathbf{z}_{n-1}, \mathbf{W}_n, \mathbf{b}_n)$, where $n \in [1, \cdots, N]$ is the

number of the actual layer. The input $\mathbf{z}_{n-1} \in \mathbb{R}^{C_{in} \times w_n \times h_n}$ is the output of the previous convolution or the network input $\mathbf{x}$, and the output feature map is $\mathbf{z}_n \in \mathbb{R}^{C_{out} \times w_{n+1} \times h_{n+1}}$, where $w_n$ and $h_n$ are the width and height, respectively. $C_{in}$ defines the number of input channels and $C_{out}$ the number of output channels. The internal parameters of the convolution consist of the weight $\mathbf{W}_n \in \mathbb{R}^{C_{out} \times C_{in} \times k \times k}$ with the kernel size $k$ and the optional bias $\mathbf{b}_n \in \mathbb{R}^{C_{out}}$. The convolution's weight $\mathbf{W}_n$ can be divided into the output channels $\mathbf{C}_{n,c} \in \mathbb{R}^{C_{in} \times k \times k}$ with $c \in 1, \ldots, C_{out}$.

Pruning aims to remove output channels of a convolution to reach a predefined model sparsity $s$, e.g., for a sparsity $s = 0.2$, we delete 20% of the channels. To specify the channels to prune, we define a sparsity vector $\mathbf{v}_n \in \{0, 1\}^{C_{out}}$ for convolution $conv_n$, such that the following condition is fulfilled

$$1 - \frac{1}{c_{out}} \sum_{c=1}^{C_{out}} \mathbf{v}_n(c) = s. \tag{1}$$

The pruning methods differ in the way the sparsity vector $\mathbf{v}_n$ is determined. The output channel $\mathbf{C}_{n,c}$ of convolution $Conv_n$ corresponding to a 0-entry in the sparsity vector $\mathbf{v}_n$ is removed during pruning. The compressed weight $\mathbf{W}_{n,r} \in \mathbb{R}^{C_{out,r} \times C_{in} \times k \times k}$ contains only the channels with a 1-entry in the sparsity vector $\mathbf{v}_n$ and has a reduced number of output channels $C_{out,r}$. Besides the weights, the bias must also be adapted, and the values corresponding to the 0-entries in $\mathbf{v}_n$ are deleted to obtain the new bias $\mathbf{b}_{n,r} \in \mathbb{R}^{C_{out,r}}$. The reduction of the output channels of $Conv_n$ also entails an adaption of the input channels of $Conv_{n+1}$. Thus, the input channels of $Conv_{n+1}$ that correspond to the 0-entries of $\mathbf{v}_n$ are deleted, and the new weight $\mathbf{W}_{n+1} \in \mathbb{R}^{C_{out} \times C_{in,r} \times k \times k}$ with the reduced number of input channels $C_{in,r}$ is obtained.

Due to the pruning, the neural network forgets some of its knowledge. Therefore, a fine-tuning step is performed after all layers in the network are pruned. In the fine-tuning step, the original training images are used to update the reduced neural network parameters with backpropagation. In the following, we present different methods to define the sparsity vector $\mathbf{v}_n$.

**Random Pruning** The easiest way to remove parameters from a neural network is to remove them randomly without considering their importance. For the random pruning, we randomly set values in the sparsity vector $\mathbf{v}_n$ to 0. The only claim that has to be fulfilled is Equation 1. The randomly defined sparsity vector $\mathbf{v}_n$ is used in the pruning to remove the channels corresponding to a 0-entry in the vector and keep them corresponding to a 1-entry, as described in the general pruning definition. All convolutions in a neural network are pruned with a unique sparsity vector $\mathbf{v}_n$.

**L1 Pruning** In random pruning, parameters with essential knowledge can be removed unintentionally, which results in a higher performance loss during pruning and more expensive fine-tuning afterward. To overcome this issue, the L1 pruning (Li et al., 2017) identifies unimportant parameters according to the L1-norm. For all channels $\mathbf{C}_{n,c}$ of weight tensor $\mathbf{W}_n$, a sensitivity measure $S_{n,c}$ is calculated by

$$S_{n,c} = \sum_{i=1}^{C_{in}} \sum_{j_1=1}^{k} \sum_{j_2=1}^{k} \mathbf{C}_{n,c}(i, j_1, j_2), \tag{2}$$

where $k$ is the size of the squared kernel and $C_{in}$ the number of input channels. The sparsity vector $\mathbf{v}_n$ is extracted from the sensitivities $\mathbf{S}_n \in \mathbb{R}^{C_{out}}$ that contains the channel sensitivities $S_{n,c}$. For the smallest sensitivities, the corresponding value in the sparsity vector is set to 0 to fulfill the condition from Equation 1. This is based on the assumption that small weights correspond to weak activation functions, which have less influence on the final result.

**Batch Normalization Pruning** If a Batch Normalization (Ioffe & Szegedy, 2015) layer follows the convolution, an alternative to the L1 pruning is Batch Normalization pruning (Liu et al., 2017). In the Batch Normalization, the output feature map of the convolution $\mathbf{z}_n$ is normalized as follows:

$$\tilde{\mathbf{z}}_n = \frac{\mathbf{z}_n - \mathrm{E}[\mathbf{z}_n]}{\sqrt{\mathrm{Var}[\mathbf{z}_n] + \epsilon}} \gamma + \beta, \tag{3}$$

where $\gamma$ is a scaling factor, $\beta$ is an offset, $\epsilon$ is a small factor to prevent zero division, and $\mathrm{E}[\mathbf{z}_n]$ stands for the expected mean of the input data with the variance $\mathrm{Var}[\mathbf{z}_n]$. The neural network learns the importance of the channels using the parameter $\gamma$ during training. Like in L1 Pruning, the assumption is that weak activation functions influence less the overall network performance. Therefore, a small $\gamma$ indicates less important channels. We convert $\gamma$ to the sparsity vector $\mathbf{v}_n$ by setting small $\gamma$-values in $\mathbf{v}_n$ to 0.

**HRank Pruning**  HRank (Lin et al., 2020) is a pruning method that determines the importance of a channel not only by the model's internal knowledge but also by considering the input data's influence. Therefore, the output feature map $\mathbf{z}_n$ produced by the weight $\mathbf{W}_n$ that should be pruned is investigated. For each channel $c_f \in 1, \dots, C_{out}$ of the features $\mathbf{z}_n$, the matrix rank is calculated with Single Value Decomposition and is averaged over a mini-batch of images. HRank shows that the usage of a mini-batch is sufficient because the rank of each channel only has a low variance between the images. The rank vector that contains the average rank for each channel can be converted into the sparsity vector $\mathbf{v}_n$. Therefore, the value in the sparsity vector $\mathbf{v}_n$ is set to 0 for a small rank, otherwise to 1. The assumption is made that if a feature map has many linear dependent rows, it has less knowledge and can be removed.

**ResRep Pruning**  Unlike the other pruning methods, ResRep (Ding et al., 2021) does not prune the model first and then fine-tune it; instead, it performs the pruning during the fine-tuning step. ResRep prunes the channels over several epochs during fine-tuning and transfers the knowledge from the pruned channels to others. Therefore, extra layers are added after the Batch Normalization, so-called compactors. These are initialized with the identity matrix so that the performance of the modified network is the same as for the original model. During training, the compactor weights for channels with less information are pushed to zero, so these channels can be pruned afterward without performance loss. The compactor's weights are pushed to zero, wherefore the gradients $\mathbf{G}_c$ are calculated as follows

$$\mathbf{G}_c = \frac{\partial \mathcal{L}}{\partial \mathbf{C}_c} v_{n,c} + \lambda \frac{\mathbf{C}_c}{||\mathbf{C}_c||_E}, \tag{4}$$

where $\lambda$ is a weighting factor. $\mathbf{G}_c$ is influenced by the standard task loss $\mathcal{L}$ (e.g., cross-entropy loss) and the Euclidean norm over the corresponding channel weights $\mathbf{C}_c$. The parameter $v_{n,c}$ is entry $c$ of mask $\mathbf{v}_n$ that is set to 0 if the channel should be pruned, else to 1. In the beginning, all values in $\mathbf{v}_n$ are set to 1, and during training, the values for channels with small weights in the compactor are set to 0 until the desired sparsity is reached. If the corresponding value $v_c$ is 1, both loss terms compete against each other. Setting $\mathbf{v}_n$ to 0 for the channels to be pruned helps to speed up the pruning, as the Euclidean norm loss does not compete with the standard loss while pushing the compactor weights to zero. After this procedure, the channels with a 0-entry in $\mathbf{v}_n$ can be pruned without further fine-tuning.

## 3.2 Uncertainty Estimation

We examine the effect of pruning on three uncertainty methods, namely the maximum softmax probability (MSP) (Hendrycks & Gimpel, 2017), Monte Carlo (MC) dropout (Gal & Ghahramani, 2015), and boot-strapped ensembles (Lakshminarayanan et al., 2017). Next, we shortly present each method.

**Maximum Softmax Probability**  The softmax function normalizes the neural network's output logits $\mathbf{z}_N$ into a probability distribution, where all values are in the range $[0, 1]$ and sum up to 1. $N$ refers to the last network layer. The definition of the softmax can be expressed for vector $\mathbf{z}_N$ with $C$ elements, one for each class, as follows:

$$\sigma(z_N)_j = \frac{\exp(z_{N,j})}{\sum_{i=1}^{C} \exp(z_{N,i})}, \quad j = 1, \dots, C. \tag{5}$$

The overall network output $\hat{\mathbf{y}}$ is the re-scaled logits $\sigma(\mathbf{z}_N)$. In image classification, the predicted class is determined by the index of the element with the maximum value in $\sigma(\mathbf{z}_N)$. The MSP (Hendrycks & Gimpel, 2017) is an easy way to assess the uncertainty of a neural network for classification. Here, the softmax value of the predicted class is used to determine the uncertainty $u = 1 - \max(\sigma(\mathbf{z}_N))$. A lower uncertainty value means a higher probability that the predicted class is correct. By implication, a high value indicates a high uncertainty.

**Bootstrapped Ensembles**  Another way to calculate the model's uncertainty is bootstrapped ensembles (Lakshminarayanan et al., 2017). This approach relies on the output of $M$ neural networks $f_m(\cdot)$ with $m \in \{1, \ldots, M\}$ to estimate the uncertainty as the variance over the $M$ predictions. Each of the $M$ models is trained with the same dataset and architecture but a different initialization as a starting point. Furthermore, the order of the data samples during the training is shuffled. These two techniques introduce randomness in the training whereby the predictions of the $M$ neural networks are expected to be equal for data observed during training and differ for data not included in the training. The overall prediction for a data sample $\mathbf{x}$ is calculated as the mean $\mu$ and the variance $\sigma^2$ over the $M$ networks by

$$\mu(\mathbf{x}) = \frac{1}{M} \sum_{m=1}^{M} f_m(\mathbf{x}), \qquad \sigma^2(\mathbf{x}) = \frac{1}{M} \sum_{m=1}^{M} (f_m(\mathbf{x}) - \mu(\mathbf{x}))^2. \tag{6}$$

The mean's softmax $\sigma(\mu(\mathbf{x}))$ depicts the overall network output $\hat{\mathbf{y}}$, and the index of the maximum entry from $\sigma(\mu(\mathbf{x}))$ is used as the class prediction. The class prediction's entry of the variance $\sigma^2(\mathbf{x})$ is associated with the uncertainty $u$, where a high variance represents a high uncertainty.

**Monte Carlo Dropout**  A main drawback in bootstrapped ensembles is the high training effort due to the $M$ independently trained neural networks. To prevent the training of multiple neural networks and approximate Bayesian neural networks, MC dropout (Gal & Ghahramani, 2015) applies the dropout layer (Srivastava et al., 2014) to bring randomness into the $M$ predictions. Therefore, the dropout layer is not only activated during training but also during inference. Because of the activated dropout during inference, the predictions of $M$ forward runs differ, and the mean $\mu(\mathbf{x})$ and variance $\sigma^2(\mathbf{x})$ can be calculated according to Equation 6. As in bootstrapped ensembles, the mean's softmax determines the network output $\hat{\mathbf{y}}$ that is used to predict the class, and the variance is associated with the uncertainty $u$.

## 4 Problem Formulation

Consider the neural network $f(\cdot)$ that is parameterized by $\theta$ and predicts the probability distribution $\hat{\mathbf{y}} = f(\mathbf{x}, \theta)$ for the input image $\mathbf{x} \in \mathbb{R}^{3 \times w \times h}$ with width $w$ and height $h$. The network is trained on the training set $\mathcal{D} = \{\mathbf{x}_i, \mathbf{y}_i\}_{i=1}^{|\mathcal{D}|}$, which contains the RGB images $\mathbf{x}$ with the corresponding labels $\mathbf{y}$, where $\mathbf{y}$ defines the one-hot class label $\mathbf{y} \in \{0, 1\}^C$ for a $C$-category classification problem, such that $\sum_{i=1}^{C} \mathbf{y}(i) = 1$. When pruning (Section 3.1) is applied to the neural network $f(\cdot)$, it affects the model's parameters $\theta$ and, therefore, its prediction $\hat{\mathbf{y}}$. After pruning, we obtain the pruned neural network $f_p(\cdot)$ with the reduced parameter set $\theta_p$. We then use well-established uncertainty estimation approaches (Section 3.2) for obtaining the uncertainty estimate $u \in \mathbb{R}$ next to the class prediction $\hat{\mathbf{y}}$. The pruning impacts not only the neural networks class prediction $\hat{\mathbf{y}}$ but also the uncertainty estimate $u$. Because of that, we investigate the influence of channel pruning on the neural network's classification performance as well as on the uncertainty estimation performance. To quantify the quality of the uncertainty estimation, a metric for the uncertainty of an image classification neural network is presented in the next section.

## 5 Proposed Uncertainty Metric

The effect of pruning on uncertainty estimation is examined with three different evaluation metrics, namely the accuracy, the expected calibration error (ECE) (Naeini et al., 2015), and the Area Under the Sparsification Error (AUSE) (Ilg et al., 2018). The metrics assess the pruned model's performance regarding the predicted class and the uncertainty prediction. The accuracy evaluates if the neural network predicts the correct class label for the input image and is therefore used for the classification evaluation. The ECE assesses the calibration of the classification neural network, i.e., it checks how well the predicted uncertainty matches the true uncertainty. The uncertainty range is divided into equal-spaced bins, and the predictions are sorted into the bins to calculate the ECE. For each bin, the mean predicted uncertainty and the true uncertainty are determined. The ECE is the weighted absolute difference between both values over all bins.

The AUSE metric was first introduced for the uncertainty evaluation in the regression task optical flow estimation (Ilg et al., 2018) to determine if the predicted uncertainty corresponds with the true uncertainty.

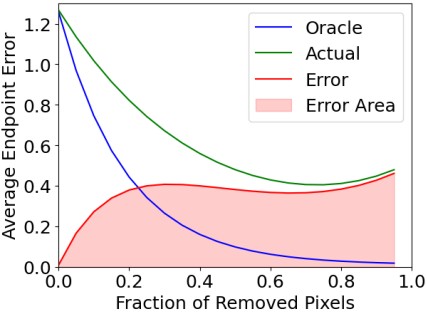
(a) AUSE for optical flow estimation

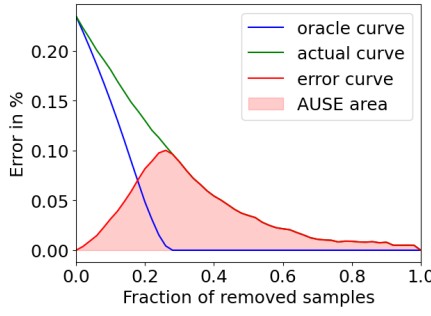
(b) AUSE for image classification

Figure 1: The AUSE (light red area) is defined as the area under the sparsification error (red). The sparsification error is the difference between the oracle curve (blue) and the sparsification curve (green). The x-axis represents the percentage of removed test set samples, and the y-axis shows the error with the corresponding error metric.

As a base for the AUSE, the oracle and actual sparsification plots are determined, represented in Fig. 1a as a blue and green curve, respectively. In optical flow estimation, the pixels with the highest uncertainty are removed step by step from the test set to generate the sparsification plot. This means that a pre-defined fraction of pixels with the currently highest uncertainty is removed, and the error metric is re-calculated to get the value of the sparsification plot for the remaining data. This is done until all pixels are removed from the test set. The error metric used in optical flow estimation is the average endpoint error. If the uncertainty corresponds with the true error, the sparsification plot decreases when removing the uncertain pixels. Removing the pixels with the true uncertainty, e.g., the prediction error, leads to the oracle sparsification plot, which determines the lower limit (blue curve in Fig 1a). In contrast, the sparsification plot decreases slower with the predicted uncertainty because of non-optimal predictions and is called the actual sparsification plot (green curve in Fig 1a). The red curve in Fig. 1a represents the difference between the two curves and the area under the sparsification error curve (represented in light red in Fig. 1a) determines the AUSE.

We use the AUSE not for uncertainty estimation in a regression task, like optical flow estimation, but for uncertainty evaluation in image classification. Instead of removing pixels during the evaluation, we remove the images with the highest classification uncertainty. Furthermore, as an error metric, we do not use the average endpoint error as in optical flow estimation but the classification error in percentage. An example plot of the AUSE for image classification is presented in Fig. 1b. The predicted uncertainty in image classification is the uncertainty of the predicted class, whereas the true uncertainty is the uncertainty of the ground truth image class. An algorithm is presented in Appendix D for a deeper understanding of the AUSE calculation for image classification.

## 6 Analysis of Pruning Effects on Uncertainty Estimation

We conduct an extensive evaluation to show the influence of pruning on the different uncertainty estimation approaches for image classification. Therefore, we combine the pruning methods from Section 3.1 with the uncertainty methods from Section 3.2. This concatenation is presented visually in Appendix A. First, we describe the experimental setup, including datasets, implementation details, and metrics. After the experimental setup, we present our results.

### 6.1 Experimental Setup

**Datasets** Our experiments use the small-scale dataset CIFAR-100 (Krizhevsky et al., 2009) and the large-scale dataset ImageNet (Deng et al., 2009). The CIFAR-100 dataset contains 50000 images for training and 10000 for evaluation while having 100 classes. The image resolution in the CIFAR-100 dataset is $32 \times 32$

pixels. ImageNet has 1000 classes and consists of over 1.2 million training images and 50000 validation images with a resolution of $224 \times 224$ pixels. The evaluation of the CIFAR-100 dataset is done on the corresponding test set, while we use the validation set for ImageNet.

**Implementation Details**    In the case of the CIFAR-100 dataset, we use the ResNet18 (He et al., 2016) and VGG19 (Simonyan & Zisserman, 2015) architectures, while we limit ourselves to ResNet18 for ImageNet. Before pruning the models, we train each model with the chosen uncertainty method. This leads to one model for MSP, one for MC dropout with dropout activated during training, and $M$ different models for bootstrapped ensembles, where we choose $M = 5$. We include a dropout layer after each block with the dropout probability set to 0.2 for MC dropout in ResNet18. In VGG19, we add the dropout layer before the channel upsampling. In order to measure uncertainty using MC dropout, five forward runs are performed. We vary the initialization, and the training data order between the training runs for bootstrapped ensembles. For random pruning, L1 pruning, Batch Normalization pruning, and HRank pruning, we directly prune all layers and perform one fine-tuning step at the end with 150 epochs for CIFAR-100 and 45 epochs for ImageNet. The pruning in ResRep is conducted during the fine-tuning, which is performed 180 epochs for both datasets. Pruning ResNet in one step enables the pruning of all layers, while during step-wise pruning, like in ResRep, only layers unaffected by the skip connections can be pruned. This is caused by the mismatch of the dimensions after the skip-connection during step-wise pruning. For all pruning techniques, we prune the models for the CIFAR-100 dataset with the sparsities 0.2, 0.4, 0.6, and 0.8. For ImageNet, we select the sparsities 0.2 and 0.6. We provide further implementation details in Appendix B.

**Metrics**    Consistent with pruning literature for image classification (Li et al., 2017; Liu et al., 2017; Lin et al., 2020; Ding et al., 2021), we report the model's prediction performance with the accuracy in percentage. We use the Area Under the Sparsification Error (AUSE) adapted for image classification to quantify the uncertainty performance, as presented in Sec. 5. Furthermore, we apply the expected calibration error (ECE) (Naeini et al., 2015) to determine the calibration quality of the pruned models. Therefore, we normalize the uncertainty prediction of MC dropout and bootstrapped ensembles into the range $[0, 1]$.

## 6.2    Experimental Results

In the following, we present the results for the different datasets and architectures. In the Figures 2, 3, and 4 the AUSE is shown in the left part (Subfigures (a)-(c)), while the ECE is plotted in the right part (Subfigures (d)-(f)). The x-axis of the plots refers to the AUSE or the ECE metric, and the y-axis to the accuracy of the classification performance. The different colors represent the five pruning methods, while the size of the marker correlates with the pruning sparsity, i.e., small markers stand for low pruning rates and large markers for high pruning rates. Additionally, the model size of the different pruning approaches and sparsities is given in Appendix C. Furthermore, we provide plots for the AUSE, ECE, and the accuracy over the sparsity and tables with accurate values for both architectures in Appendix E and F.

**CIFAR-100**    The outcomes of the experiments on CIFAR-100 for the ResNet18 architecture are presented in Figure 2 and for the VGG19 architecture in Figure 3. The results of the ResNet18 model show that for all uncertainty and pruning methods, the AUSE, like the accuracy, decreases when increasing the pruning sparsity. A comparison of the pruning methods reveals that all pruning methods, except ResRep pruning, have similar degradation of AUSE and accuracy for all uncertainty methods. In the case of the MSP uncertainty, the ResRep approach behaves similarly to the other pruning methods for low pruning rates but decreases more with higher pruning rates. For bootstrapped ensembles and MC dropout, the degradation of the uncertainty measure for ResRep pruning is faster. The degeneration of the accuracy near to random with MC dropout during ResRep pruning with a pruning sparsity of 0.8 can be explained by the fact that the model converges slower with the inserted dropout layer than without this layer, and we keep the training epochs constant. The main difference between the performance of the uncertainty methods is in the absolute AUSE values. Here, each uncertainty method starts with a different performance before the pruning and worsens to different degrees.

Besides the AUSE, we use the ECE to evaluate the pruning's influence on the uncertainty prediction performance and show the accuracy over the ECE in the right part of Figures 2 and 3. The results for the ResNet18

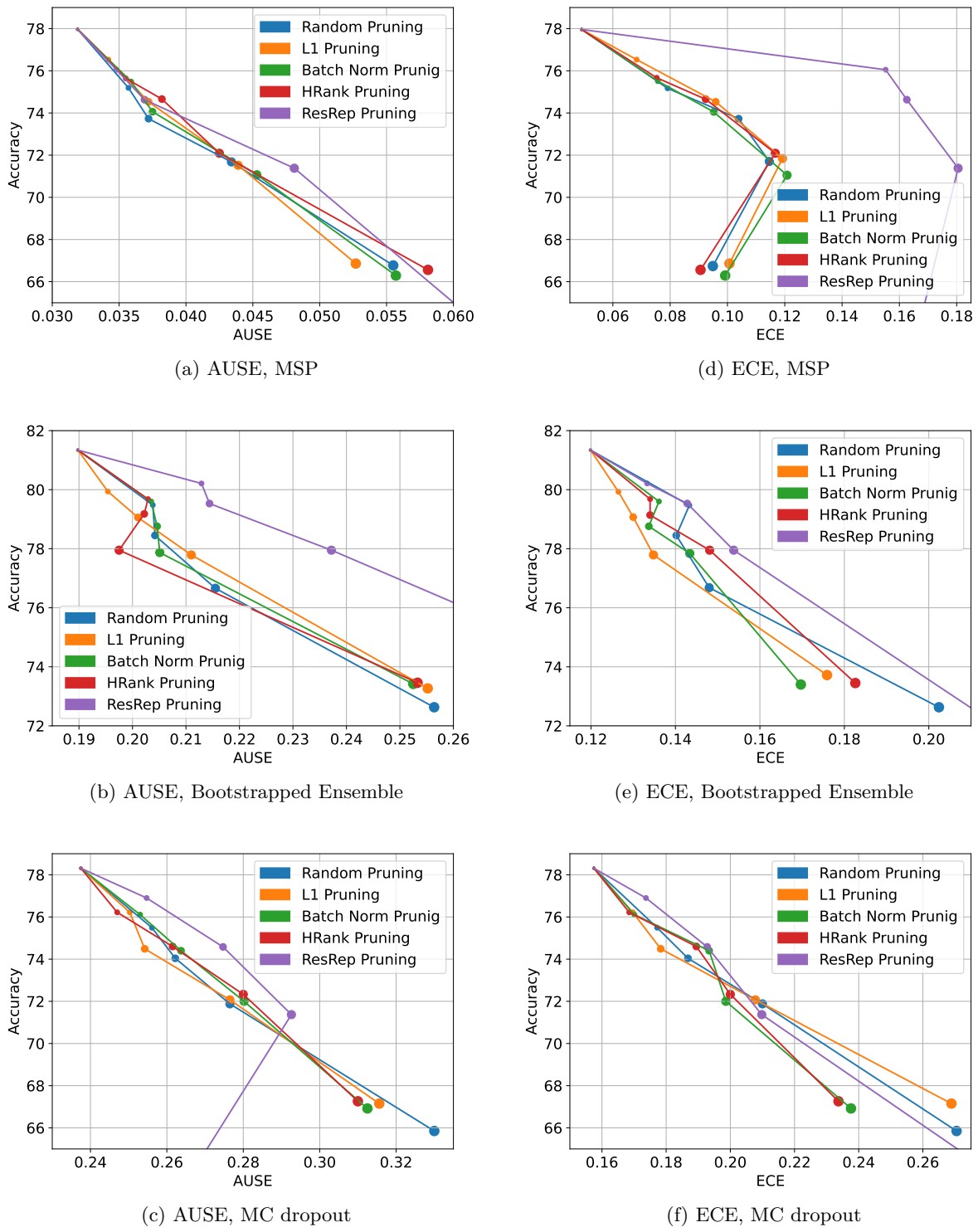

Figure 2: CIFAR-100 results for ResNet18 with the AUSE in (a)-(c) and with the ECE in (d)-(f). The accuracy is plotted over the AUSE or ECE for the three uncertainty methods MSP, bootstrapped ensembles, and MC dropout. The marker size displays the pruning sparsities, i.e., small markers denote low and large markers high pruning rates.

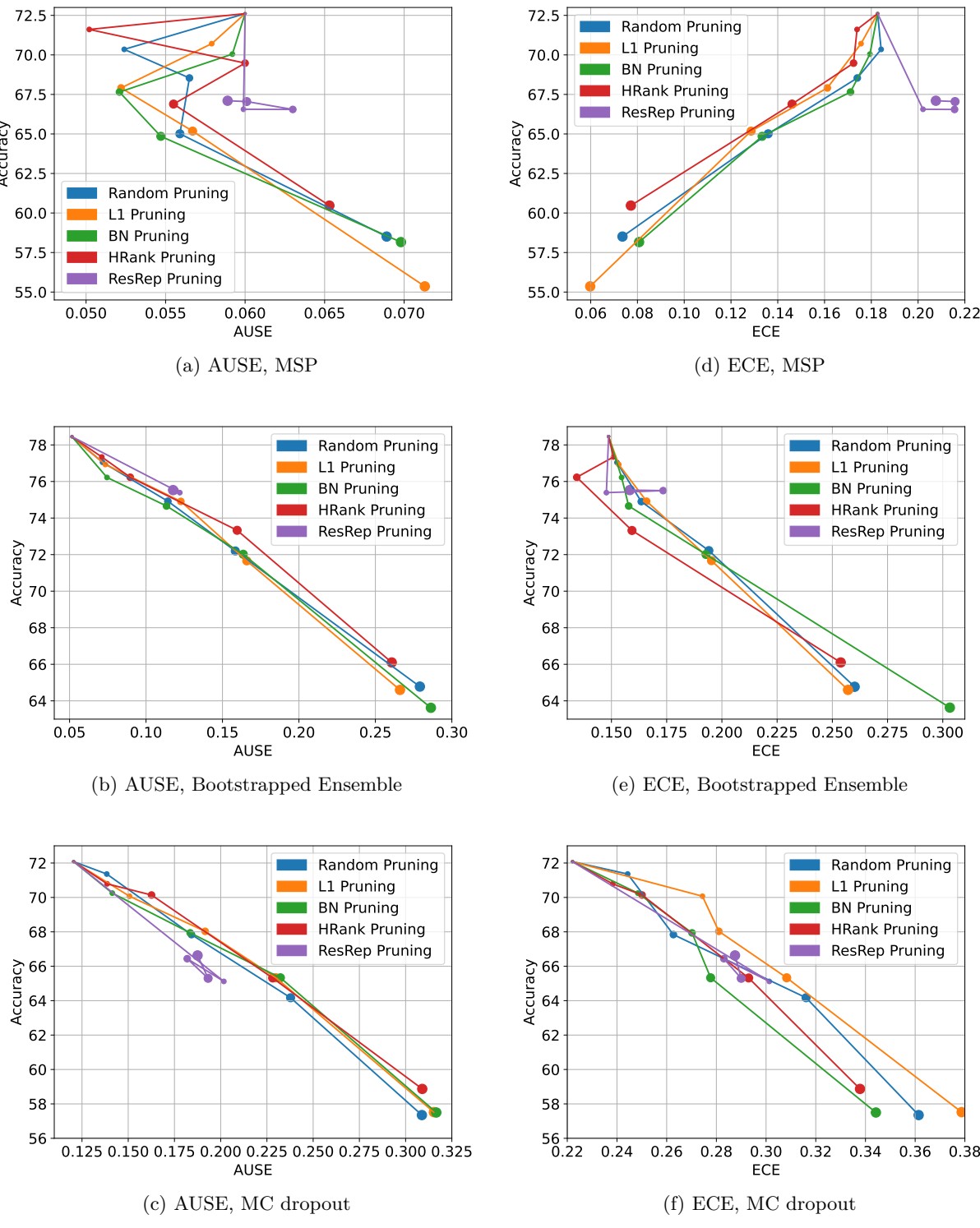

Figure 3: CIFAR-100 results for VGG19 with the AUSE in (a)-(c) and with the ECE in (d)-(f). The accuracy is plotted over the AUSE or ECE for the three uncertainty methods MSP, bootstrapped ensembles, and MC dropout. The marker size displays the pruning sparsities, i.e., small markers denote low and large markers high pruning rates.

model are presented in Figure 2. As for the AUSE, the curves proceed similarly within each uncertainty method for the random, L1, BN, and HRank pruning. The course of the ResRep pruning is also similar, but the values differ from the other methods. Furthermore, the ECE worsens during the pruning and improves at a sparsity of 0.8 for the MSP. In contrast, the other uncertainty methods (bootstrapped ensembles and MC dropout) decreases the ECE for each pruning sparsity. A ResRep-specific detail is that the accuracy drops quickly with a pruning sparsity of 0.8. The performance of VGG19 trained on CIFAR-100 is presented in Figure 3. The overall findings for VGG19 are comparable to the results from ResNet18. The AUSE and accuracy become worse with higher pruning rates. For the MSP, the AUSE for VGG19 is higher than for ResNet18, while the uncertainty of the unpruned model is lower for the other uncertainty methods. Notably, for the MSP, the pruning sparsities 0.2, 0.4, and 0.6 have a better uncertainty performance (AUSE) than the original model, and only the performance of sparsity 0.8 is worse than the performance of the original model. Furthermore, the performance of ResRep drops with the pruning sparsity 0.2 but stays constant after the drop for the following sparsities.

The right image side of Figure 3 shows the ECE results with the VGG19 backbone. The ECE behaves equally for the random, L1, BN, and HRank pruning method. The MSP improves the neural network calibration for higher pruning sparsities, as the ECE shows, but the accuracy diminishes with the higher sparsities. Using bootstrapped ensembles or MC dropout decreases the ECE with a higher pruning rate. In contrast to these four pruning methods, the ECE performance of the ResRep pruning is more constant over all pruning sparsities.

**ImageNet**   To prove that our findings are not only valid for small-scale datasets, we evaluate the effect of the pruning methods on the uncertainty prediction for ImageNet. We show the AUSE results of this experiment performed with ResNet18 on the left side of Figure 4. The general finding is that the AUSE and the accuracy decrease with a higher pruning sparsity, following the conclusion made for the CIFAR-100 dataset. Compared to the small-scale dataset, the overall values for the AUSE are higher, which means that the uncertainty prediction worsens. As emerged on the CIFAR-100 dataset, the evaluation on ImageNet confirms that the AUSE and accuracy are correlated, i.e., a lower accuracy also leads to a lower AUSE. There is little variance between the pruning methods on ImageNet, and all curves show a similar behavior. An exception to this is ResRep, which has a superior performance compared to the others but also leads to slightly more parameters after pruning (see Appendix C). ResRep also proves that better accuracy leads to an improved AUSE prediction. For bootstrapped ensembles, the model pruned with a sparsity of 0.2 is even better than the unpruned baseline. Further, in Figure 4c, we can see that with a sparsity of 0.6, the dropout layer removes too much information, leading to a decreased accuracy.

The ECE presents further insights into the influence of pruning on the neural network's uncertainty prediction, which we show on the right side in Figure 4. The behavior of the ECE for the MSP, the bootstrapped ensembles, and the MC dropout uncertainty methods is comparable with the CIFAR-100 dataset for ResNet18. The accuracy, as well as the ECE, decreases with a higher pruning sparsity. An exception is the ResRep pruning combined with bootstrapped ensembles, where the accuracy improves for a pruning rate of 0.2. Furthermore, the ResRep pruning diminishes the accuracy and the ECE less than the other pruning methods. This could be caused by the higher number of parameters after the ResRep pruning.

## 7   Conclusion

In this work, we evaluate the influence of pruning on the uncertainty estimation of a neural network for image classification and, therefore, combine different uncertainty estimation approaches with pruning methods. We conduct our experiments on CIFAR-100 and ImageNet with the ResNet18 and VGG19 architectures. For the uncertainty quantification, we propose the Area Under the Sparsification Error (AUSE) metric for image classification and apply the excepted calibration error (ECE). The evaluation's main findings are that pruning affects the class prediction and the uncertainty estimation in a negative way. Furthermore, the tested pruning methods influence the uncertainty estimation methods similarly. Improving the pruning approach w.r.t the accuracy leads to a better uncertainty estimation. Finally, the pruning'S influence on the MSP is smaller than on the other uncertainty approaches, and for VGG19, small pruning sparsities improve the AUSE. A more detailed look at the ECE shows that pruning can enhance model calibration when using the MSP.

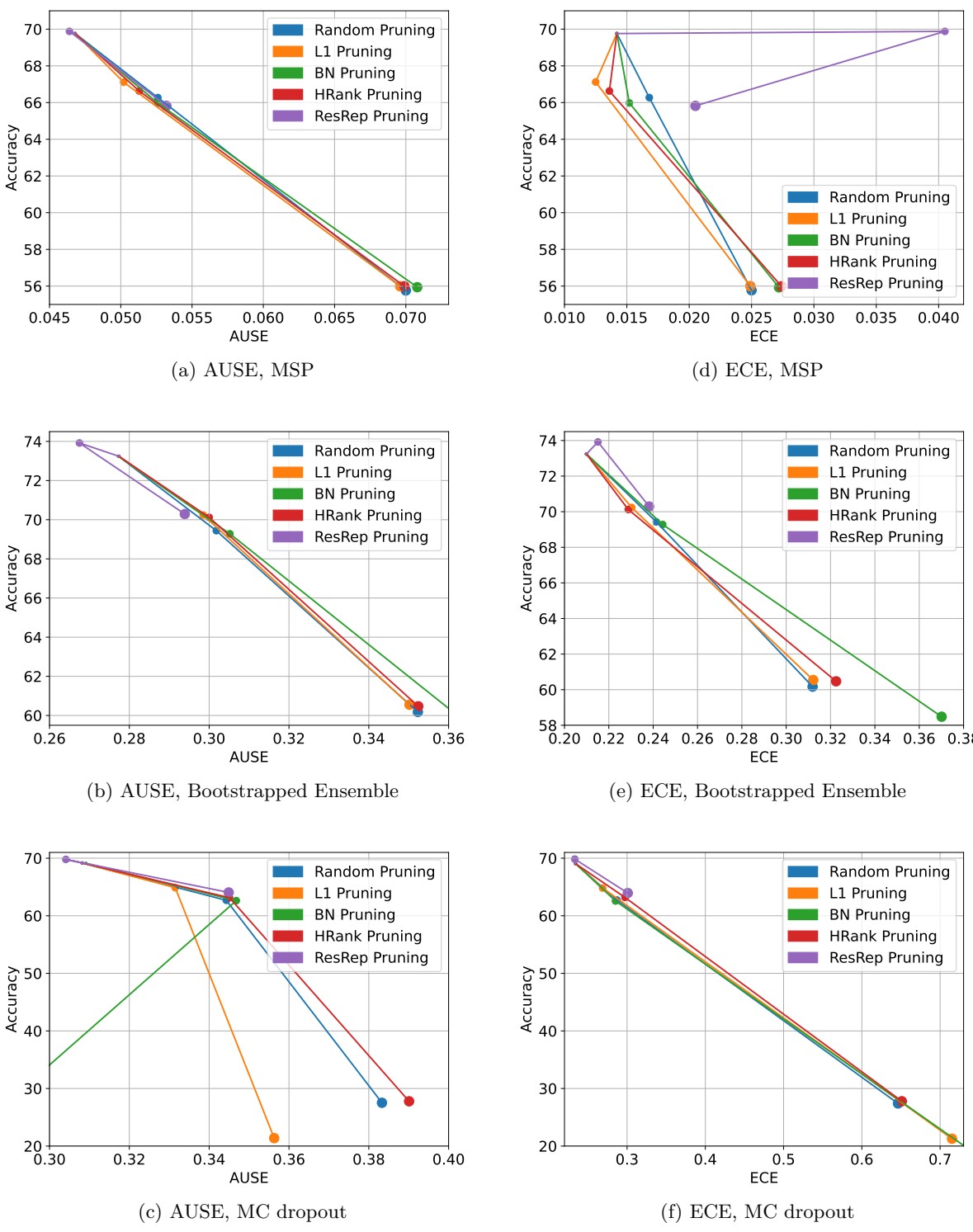

Figure 4: ImageNet results for ResNet18 with the AUSE in (a)-(c) and with the ECE in (d)-(f). The accuracy is plotted over the AUSE or ECE for the three uncertainty methods MSP, bootstrapped ensembles, and MC dropout. The marker size displays the pruning sparsities, i.e., small markers denote low and large markers high pruning rates.

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

# A    Pruning and Uncertainty Method Combination

Fig. 5 illustrates the combination of the pruning methods with the three uncertainty techniques MSP, bootstrapped ensembles, and MC dropout.

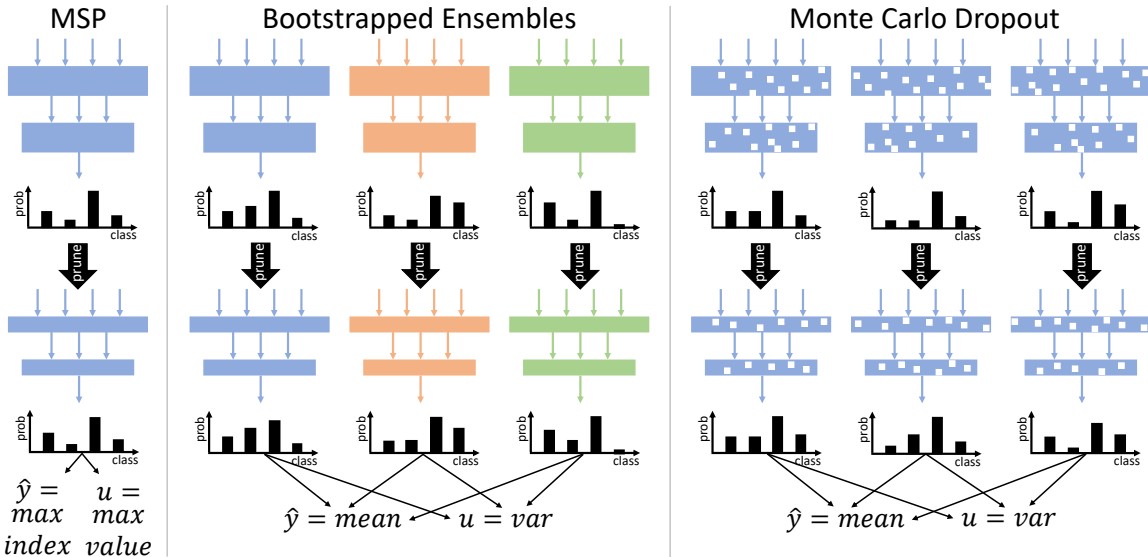

Figure 5: The image shows how we combine the pruning methods with the uncertainty estimation techniques. In the maximum softmax probability (MSP), we use the index of the maximum value as class prediction and its value as an uncertainty measure. The different colors of the layers in bootstrapped ensembles define the different parameter sets. In dropout, the white squares in the layers depict the omitted parameters by the activated dropout. For bootstrapped ensembles and MC dropout, we calculate the mean and variance and use the index of the maximum as the predicted class and the variance as an uncertainty measure.

# B    Implementation Details

In the following, we explain in detail what code we use as a base for our experiments. On the CIFAR100 dataset, we train ResNet18 He et al. (2016) and VGG19 Simonyan & Zisserman (2015), where we build on the public implementation of pytorch-cifar[1]. As a base for the training of the ResNet18 on ImageNet, we use the official Pytorch example[2]. In both cases, we re-use the training hyperparameters from the base code for fine-tuning. We implement random pruning, L1 pruning Li et al. (2017), and Batch Normalization pruning Liu et al. (2017) according to their papers. For HRank pruning[3] and ResRep pruning[4], we use their official code as a base. To determine the maximum softmax probability, we normalize the output logits of the last network layer with a standard softmax. MC dropout Gal & Ghahramani (2015) and bootstrapped ensembles Lakshminarayanan et al. (2017) are built on the description of their papers.

# C    Neural Network Size

The evaluation of the model size is presented in Table 1 for the ResNet18 and VGG19 architecture. For both architectures, the model size for Random pruning, L1 Pruning, Batch Normalization Pruning, and HRank Pruning is the same, while the number of parameters differs for the ResRep pruning. The identical amount of parameters is caused by the fact that the removal of the parameters is equivalent, and only the rule diverges which parameters are removed. The higher amount of parameters in VGG models pruned with ResRep

---

[1]https://github.com/kuangliu/pytorch-cifar
[2]https://github.com/pytorch/examples/blob/7ec911c46c00ac98d3adfd996ee2a011bbb9fdba/imagenet/main.py
[3]https://github.com/lmbxmu/HRank
[4]https://github.com/DingXiaoH/ResRep

stems from the channel pruning definition of ResRep. In the other pruning methods, the same percentual amount of channels is pruned in each layer. In contrast, in ResRep, the amount of pruned channels can differ between layers and have only to fulfill the sparsity over the whole neural network. Additionally, in ResNet18 models, the skip connections in the ResNet architecture hinder the pruning of the layers before the skip connection in the step-by-step pruning procedure, which increases the remaining parameters in ResRep pruning.

Table 1: Number of parameters multiplied by $1e^6$ for different sparsities (0.0, 0.2, 0.4, 0.6, 0.8) for the ResNet18 and VGG19 model.

| Model | Method/Sparsity | 0.0 | 0.2 | 0.4 | 0.6 | 0.8 |
|-------|-----------------|-----|-----|-----|-----|-----|
| ResNet18 | Random, L1, Batch Norm, HRank Pruning | 11.17 | 7.18 | 4.05 | 1.80 | 0.46 |
| | ResRep Pruning | 11.17 | 8.05 | 5.44 | 3.29 | 1.71 |
| VGG19 | Random, L1, Batch Norm, HRank Pruning | 20.04 | 12.86 | 7.26 | 3.22 | 0.82 |
| | ResRep Pruning | 20.04 | 12.41 | 7.35 | 3.78 | 3.36 |

## D   Uncertainty Metric Algorithm

Further insights into the AUSE calculation are given in Alg. 1, where the input is the predicted softmax probabilities $\hat{\mathbf{y}}$ for all classes $C$ of $T$ test set samples and the ground truth label $\mathbf{y}$ for all $T$ test set samples. In the case of the sparsification curve, the predicted uncertainty is utilized as the uncertainty measure, which is the predicted class probability subtracted from one, as shown in Alg. 1, Line 13. This means, for example, that if a sample is classified with a probability of 0.8 as a *car*, the predicted uncertainty is 0.2. The oracle sparsification curve uses the true error as uncertainty, defined by calculating the error for the ground truth class as shown in Alg. 1, Line 14. It can be distinguished between correct and miss-classified samples when determining the true error. The true error is equal to the predicted uncertainty for correct classified samples, e.g., 0.2 for our example with the *car*. Whereas for a miss-classified sample, the predicted uncertainty of the true class is used. If the true class has a probability of 0.15 and is *truck* instead of *car*, the true error is 0.85. A perfect uncertainty prediction results in equal graphs for the sparsification curve and the oracle sparsification curve, which leads to an area between both curves equal to zero. Therefore, lower AUSE values represent a better uncertainty performance. The AUSE is calculated in Alg. 1, Line 17 as the area under the curve (AUC) between both curves.

---

**Algorithm 1:** Overview of the calculation of the AUSE.

    **Input:** predicted (mean) softmax probabilities $\hat{\mathbf{y}} \in \mathbb{R}^{N \times C}$, ground truth labels $\mathbf{y} \in \mathbb{R}^N$

    **Output:** AUSE metric

**1** **def** error($\mathbf{p}$, $\mathbf{y}$, $s$):

**2**      $\mathbf{y}_s, \mathbf{p}_s = \mathbf{y}[s], \mathbf{p}[s]$                  `// Mask uncertain samples`

**3**      $a = \frac{sum(\mathbf{p}_s == \mathbf{y}_s)}{len(\mathbf{y}_s)}$

**4**      **return** $1 - a$

**5**

**6** **def** curve($\mathbf{u}$, $\mathbf{p}$, $\mathbf{y}$):

**7**      $T = [\mathbf{foreach}\ q \in range(0, 100, 2)\ \mathbf{do}\ \mathrm{percentile}_q(\mathbf{u})]$      `// Get thresholds for percentiles`

**8**      $S = [\mathbf{foreach}\ t \in T\ \mathbf{do}\ \mathbf{u} >= t]$                `// Generate percentile subsets`

**9**      $C = [\mathbf{foreach}\ s \in S\ \mathbf{do}\ \mathrm{error}(\mathbf{p}, \mathbf{y}, s)]$        `// Calculate error for each percentile`

**10**      **return** $C$

**11**

**12** $\mathbf{p} = \mathrm{argmax}(\hat{\mathbf{y}}, \dim = 1)$                `// Generate predicted classes`

**13** $u_{pred} = 1 - \max(\hat{\mathbf{y}}, \dim = 1)$            `// Generate predicted uncertainty`

**14** $u_{true} = 1 - \hat{\mathbf{y}}[:, \mathbf{y}]$                 `// Generate true uncertainty`

**15** $C_{pred} = \mathrm{curve}(u_{pred}, \mathbf{p}, \mathbf{y})$             `// Get predicted curve`

**16** $C_{true} = \mathrm{curve}(u_{true}, \mathbf{p}, \mathbf{y})$               `// Get true curve`

**17** $\mathrm{AUSE} = \mathrm{AUC}(C_{pred} - C_{true})$          `// Calculate Area Under the Curve`

---

## E   CIFAR100

In the paper, we present in the figures for CIFAR100 the accuracy plotted over the AUSE and the ECE metric. Additionally, we show in the appendix the AUSE, ECE, and accuracy over the sparsity in Figure 6. The results for ResNet18 are displayed on the left side and for VGG19 on the right side. The overall effect is that the accuracy and the AUSE get worse with a higher pruning sparsity, whereas the ECE improves in some cases. As displayed for ResNet18, the accuracy for the maximum softmax probability and MC dropout behaves similarly. For the bootstrapped ensemble, the accuracy is higher than for the other approaches. Also, the AUSE is the best for the maximum softmax probability, followed by bootstrapped ensembles and MC dropout. For VGG19, the behavior of the different pruning and uncertainty methods is similar to that for ResNet18. Comparing the ECE of both architectures shows that pruning imporves the ECE for VGG19 but not for ResNet18. For the MSP, the ECE gets worse for ResNet18 and enhances for VGG19. A more detailed look is provided as pure numbers for the AUSE in Table 2 for the ResNet18 architecture and in Table 3 for the VGG19 architecture. The numbers for the ECE are given in Table 4 for ResNet18 and for VGG19 in Table 5.

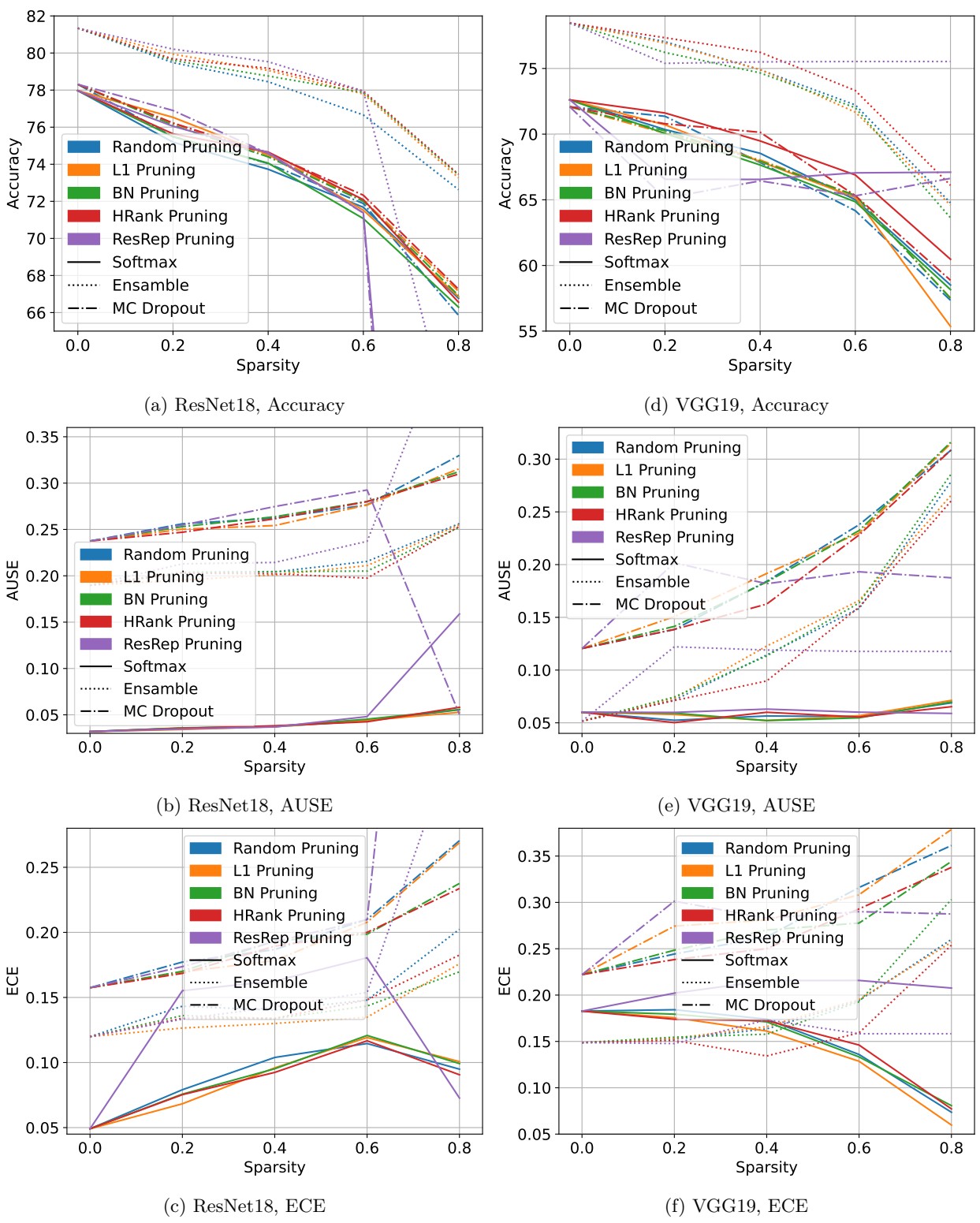

Figure 6: CIFAR100 results for ResNet18 on the left side and VGG19 on the right side with the accuracy (Subfigure 6a and 6d), the AUSE (Subfigure 6b and 6e), and the ECE (Subfigure 6c and 6f). The pruning sparsities refer to the number of pruned layers.

Table 2: CIFAR100 results for ResNet18 with the accuracy and AUSE metric (accuracy/AUSE). The pruning sparsities refer to the number of pruned layers, and the pruned models are fine-tuned with 150 epochs. The AUSE metric is scaled by $1e^{-1}$.

| Pruning Method | Uncertainty Method | Sparsities | | | | |
|---|---|---|---|---|---|---|
| | | 0.0 | 0.2 | 0.4 | 0.6 | 0.8 |
| Random Pruning | Softmax | 77.97/0.319 | 75.19/0.357 | 73.73/0.372 | 71.68/0.434 | 66.77/0.555 |
| | Ensemble | 81.34/1.898 | 79.48/2.038 | 78.45/2.042 | 76.66/2.155 | 72.63/2.564 |
| | MC Dropout | 78.31/2.375 | 75.49/2.561 | 74.04/2.622 | 71.88/2.765 | 65.85/3.300 |
| L1 Pruning | Softmax | 77.97/0.319 | 76.53/0.342 | 74.53/0.372 | 71.52/0.439 | 66.86/0.527 |
| | Ensemble | 81.34/1.898 | 79.94/1.954 | 79.06/2.010 | 77.79/2.110 | 73.27/2.552 |
| | MC Dropout | 78.31/2.375 | 76.22/2.503 | 74.49/2.542 | 72.07/2.765 | 67.15/3.156 |
| Batch Norm Pruning | Softmax | 77.97/0.319 | 75.49/0.359 | 74.06/0.375 | 71.07/0.453 | 66.29/0.557 |
| | Ensemble | 81.34/1.898 | 79.60/2.035 | 78.76/2.046 | 77.87/2.051 | 73.42/2.525 |
| | MC Dropout | 78.31/2.375 | 76.11/2.530 | 74.40/2.637 | 72.00/2.802 | 66.92/3.125 |
| HRank Pruning | Softmax | 77.97/0.319 | 75.64/0.355 | 74.66/0.382 | 72.09/0.425 | 66.56/0.581 |
| | Ensemble | 81.34/1.898 | 79.68/2.029 | 79.18/2.022 | 77.95/1.975 | 73.46/2.533 |
| | MC Dropout | 78.31/2.375 | 76.22/2.470 | 74.60/2.615 | 72.33/2.800 | 67.26/3.100 |
| ResRep Pruning | Softmax | 77.96/0.319 | 76.05/0.348 | 74.63/0.369 | 71.38/0.481 | 12.14/1.588 |
| | Ensemble | 81.34/1.898 | 80.21/2.129 | 79.53/2.144 | 77.95/2.372 | 58.38/4.891 |
| | MC Dropout | 78.31/2.375 | 76.90/2.547 | 74.58/2.747 | 71.37/2.926 | 01.54/0.504 |

Table 3: CIFAR100 results for VGG19 with the accuracy and AUSE metric (accuracy/AUSE). The pruning sparsities refer to the number of pruned layers, and the pruned models are fine-tuned with 150 epochs. The AUSE metric is scaled by $1e^{-1}$.

| Pruning Method | Uncertainty Method | Sparsities | | | | |
|---|---|---|---|---|---|---|
| | | 0.0 | 0.2 | 0.4 | 0.6 | 0.8 |
| Random Pruning | Softmax | 72.61/0.600 | 70.35/0.524 | 68.54/0.565 | 65.01/0.559 | 58.51/0.689 |
| | Ensemble | 78.45/0.516 | 77.05/0.716 | 74.92/1.143 | 72.20/1.584 | 64.78/2.791 |
| | MC Dropout | 72.08/1.205 | 71.36/1.384 | 67.84/1.842 | 64.17/2.378 | 57.35/3.088 |
| L1 Pruning | Softmax | 72.61/0.600 | 70.71/0.579 | 67.91/0.522 | 65.18/0.567 | 55.36/0.713 |
| | Ensemble | 78.45/0.516 | 76.93/0.734 | 74.91/1.228 | 71.66/1.659 | 64.60/2.661 |
| | MC Dropout | 72.08/1.205 | 70.07/1.507 | 68.03/1.916 | 65.33/2.299 | 57.52/3.153 |
| Batch Norm Pruning | Softmax | 72.61/0.600 | 70.05/0.592 | 67.65/0.521 | 64.85/0.547 | 58.16/0.698 |
| | Ensemble | 78.45/0.516 | 76.22/0.745 | 74.66/1.134 | 72.02/1.636 | 63.62/2.863 |
| | MC Dropout | 72.08/1.205 | 70.25/1.414 | 67.94/1.835 | 65.33/2.323 | 57.50/3.166 |
| HRank Pruning | Softmax | 72.61/0.600 | 71.61/0.502 | 69.48/0.600 | 66.89/0.555 | 60.47/0.653 |
| | Ensemble | 78.45/0.516 | 77.34/0.711 | 76.23/0.897 | 73.33/1.596 | 66.10/2.607 |
| | MC Dropout | 72.08/1.205 | 70.79/1.386 | 70.14/1.626 | 65.32/2.281 | 58.87/3.090 |
| ResRep Pruning | Softmax | 72.61/0.600 | 66.56/0.599 | 66.55/0.630 | 67.05/0.601 | 67.10/0.589 |
| | Ensemble | 78.45/0.516 | 75.39/1.221 | 75.50/1.190 | 75.53/1.177 | 75.53/1.177 |
| | MC Dropout | 72.08/1.205 | 65.12/2.017 | 66.44/1.819 | 65.30/1.932 | 66.63/1.875 |

Table 4: CIFAR100 results for ResNet18 with the accuracy and ECE metric (accuracy/ECE). The pruning sparsities refer to the number of pruned layers, and the pruned models are fine-tuned with 150 epochs. The ECE metric is scaled by $1e^{-1}$.

| Pruning Method | Uncertainty Method | Sparsities | | | | |
|---|---|---|---|---|---|---|
| | | 0.0 | 0.2 | 0.4 | 0.6 | 0.8 |
| Random Pruning | Softmax | 77.96/0.491 | 75.17/0.791 | 73.73/1.039 | 71.70/1.146 | 66.74/0.949 |
| | Ensemble | 81.34/1.199 | 79.47/1.433 | 78.45/1.402 | 76.68/1.480 | 72.63/2.024 |
| | MC Dropout | 78.31/1.575 | 75.49/1.772 | 74.04/1.868 | 71.88/2.100 | 65.85/2.705 |
| L1 Pruning | Softmax | 77.96/0.491 | 76.53/0.683 | 74.53/0.959 | 71.84/1.192 | 66.86/1.007 |
| | Ensemble | 81.34/1.199 | 79.93/1.265 | 79.07/1.300 | 77.79/1.348 | 73.72/1.759 |
| | MC Dropout | 78.31/1.575 | 76.22/1.696 | 74.49/1.783 | 72.07/2.078 | 67.15/2.689 |
| Batch Norm Pruning | Softmax | 77.96/0.491 | 75.51/0.757 | 74.05/0.952 | 71.05/1.208 | 66.29/0.992 |
| | Ensemble | 81.34/1.199 | 79.60/1.361 | 78.76/1.337 | 77.84/1.434 | 73.40/1.697 |
| | MC Dropout | 78.31/1.575 | 76.11/1.701 | 74.40/1.934 | 72.00/1.986 | 66.92/2.376 |
| HRank Pruning | Softmax | 77.96/0.491 | 75.67/0.753 | 74.65/0.924 | 72.09/1.167 | 66.56/0.906 |
| | Ensemble | 81.34/1.199 | 79.68/1.340 | 79.14/1.340 | 77.95/1.481 | 73.45/1.826 |
| | MC Dropout | 78.31/1.575 | 76.22/1.685 | 74.60/1.894 | 72.33/2.000 | 67.26/2.337 |
| ResRep Pruning | Softmax | 77.96/0.491 | 76.05/1.552 | 74.63/1.626 | 71.38/1.805 | 12.14/0.727 |
| | Ensemble | 81.34/1.199 | 80.21/1.333 | 79.53/1.428 | 77.95/1.538 | 58.38/3.597 |
| | MC Dropout | 78.31/1.575 | 76.90/1.737 | 74.58/1.928 | 71.37/2.098 | 01.54/8.779 |

Table 5: CIFAR100 results for VGG19 with the accuracy and ECE metric (accuracy/ECE). The pruning sparsities refer to the number of pruned layers, and the pruned models are fine-tuned with 150 epochs. The ECE metric is scaled by $1e^{-1}$.

| Pruning Method | Uncertainty Method | Sparsities | | | | |
|---|---|---|---|---|---|---|
| | | 0.0 | 0.2 | 0.4 | 0.6 | 0.8 |
| Random Pruning | Softmax | 72.61/1.827 | 70.35/1.842 | 68.54/1.740 | 65.01/1.359 | 58.51/0.736 |
| | Ensemble | 78.46/1.487 | 77.04/1.525 | 74.90/1.635 | 72.22/1.941 | 64.77/2.601 |
| | MC Dropout | 72.08/2.221 | 71.36/2.443 | 67.84/2.627 | 64.17/3.161 | 57.35/3.614 |
| L1 Pruning | Softmax | 72.61/1.827 | 70.71/1.756 | 67.91/1.612 | 65.18/1.286 | 55.36/0.598 |
| | Ensemble | 78.46/1.487 | 76.93/1.533 | 74.92/1.659 | 71.67/1.953 | 64.60/2.571 |
| | MC Dropout | 72.08/2.221 | 70.07/2.744 | 68.03/2.810 | 65.33/3.083 | 57.52/3.787 |
| Batch Norm Pruning | Softmax | 72.61/1.827 | 70.05/1.794 | 67.65/1.711 | 64.85/1.333 | 58.16/0.807 |
| | Ensemble | 78.46/1.487 | 76.22/1.546 | 74.66/1.578 | 72.01/1.928 | 63.63/3.032 |
| | MC Dropout | 72.08/2.221 | 70.25/2.484 | 67.94/2.702 | 65.33/2.777 | 57.50/3.442 |
| HRank Pruning | Softmax | 72.61/1.827 | 71.61/1.739 | 69.48/1.724 | 66.89/1.461 | 60.47/0.772 |
| | Ensemble | 78.46/1.487 | 77.34/1.509 | 76.23/1.343 | 73.32/1.593 | 66.10/2.538 |
| | MC Dropout | 72.08/2.221 | 70.79/2.383 | 70.14/2.501 | 65.32/2.930 | 58.87/3.378 |
| ResRep Pruning | Softmax | 72.61/1.827 | 66.56/2.021 | 66.55/2.156 | 67.05/2.158 | 67.10/2.076 |
| | Ensemble | 78.46/1.487 | 75.39/1.477 | 75.50/1.734 | 75.53/1.583 | 75.53/1.583 |
| | MC Dropout | 72.08/2.221 | 65.12/3.013 | 66.44/2.830 | 65.30/2.900 | 66.63/2.875 |

# F    ImageNet

For the large-scale evaluation on ImageNet, we provide additional information in Figure 7, Table 6, and Table 7. The accuracy, AUSE, and ECE over the sparsity are shown in Figure 7. It can be seen in Subfigure 7a that the accuracy drops for MC dropout with a pruning sparsity of 0.6 to around 20%. This could be due to the slower convergence when using dropout layers during training. The huge gap between the MC dropout accuracy and the accuracies of the other uncertainty methods also reflects in the ECE, where MC dropout also gives the worst performance. In contrast, the performance of the MC dropout's AUSE is comparable with those of bootstrapped ensembles. Furthermore, we can see that the MSP has the best results for the AUSE and the ECE, whereas the ECE for the MC dropout improves with higher pruning rates for BN pruning. Detailed values for the AUSE are given in Table 6 and for the ECE in Table 7.

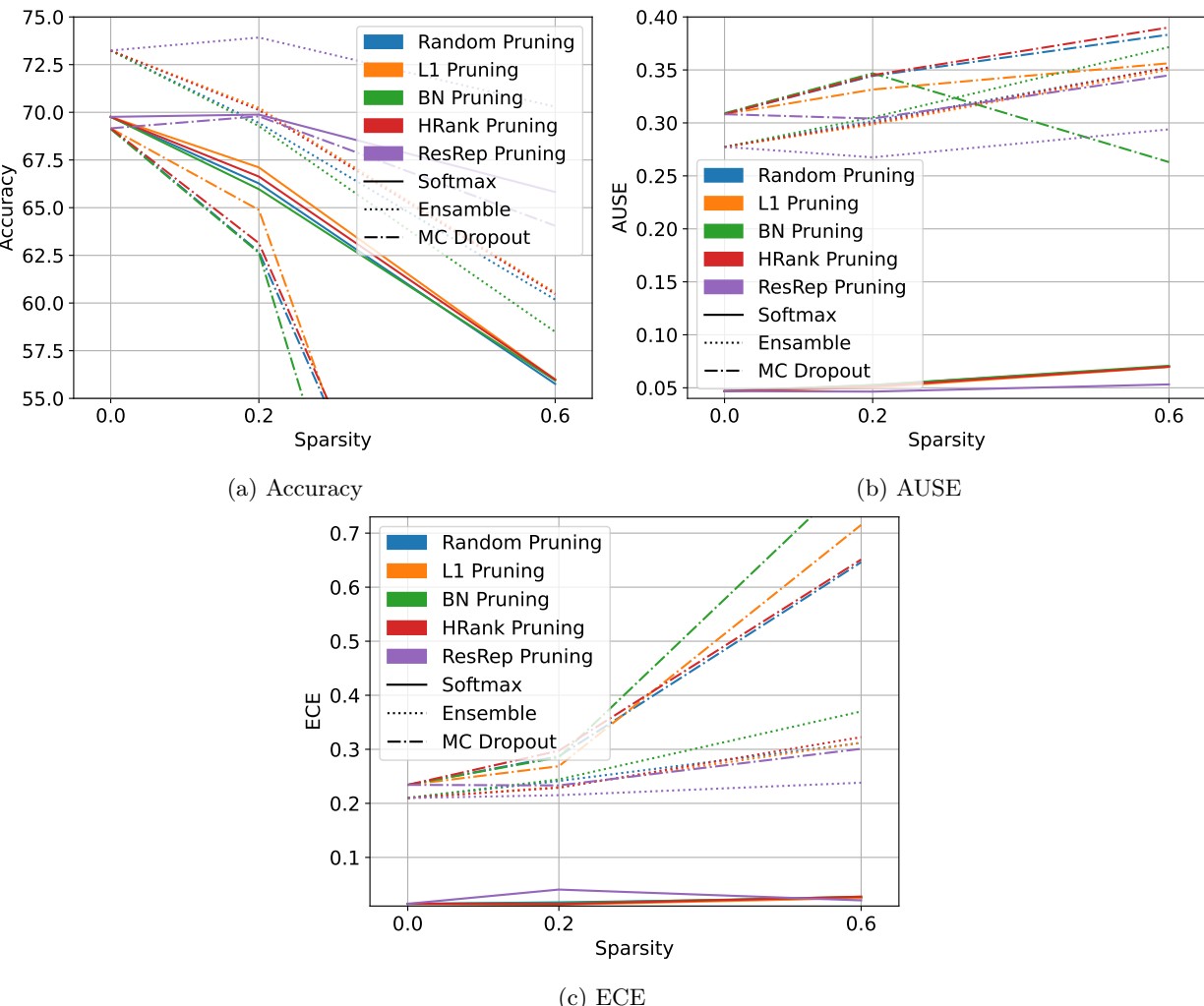

Figure 7: ImageNet results for ResNet18 with the accuracy (Subfigure 7a), the AUSE (Subfigure 7b), and the ECE (Subfigure 7c). The pruning sparsities refer to the number of pruned layers.

Table 6: ImageNet results for ResNet18 with the accuracy and AUSE metric (accuracy/AUSE). The pruning sparsities refer to the number of pruned layers, and the pruned models are fine-tuned with 150 epochs. The AUSE metric is scaled by $1e^{-1}$.

| Pruning Method | Uncertainty Method | Sparsities | | |
|---|---|---|---|---|
| | | 0.0 | 0.2 | 0.6 |
| Random Pruning | Softmax | 69.76/0.468 | 66.27/0.526 | 55.76/0.700 |
| | Ensemble | 73.24/2.773 | 69.44/3.018 | 60.18/3.523 |
| | MC Dropout | 69.16/3.082 | 62.69/3.443 | 27.52/3.833 |
| L1 Pruning | Softmax | 69.76/0.468 | 67.11/0.502 | 55.99/0.696 |
| | Ensemble | 73.24/2.773 | 70.24/2.985 | 60.55/3.502 |
| | MC Dropout | 69.16/3.082 | 64.88/3.315 | 21.38/3.563 |
| Batch Norm Pruning | Softmax | 69.76/0.468 | 65.97/0.526 | 55.94/0.708 |
| | Ensemble | 73.24/2.773 | 69.28/3.052 | 58.49/3.715 |
| | MC Dropout | 69.12/3.091 | 62.63/3.468 | 11.52/2.630 |
| HRank Pruning | Softmax | 69.76/0.468 | 66.63/0.513 | 55.99/0.699 |
| | Ensemble | 73.24/2.774 | 70.12/3.000 | 60.47/3.524 |
| | MC Dropout | 69.16/3.082 | 63.14/3.451 | 27.78/3.901 |
| ResRep Pruning | Softmax | 69.76/0.468 | 69.88/0.464 | 65.82/0.532 |
| | Ensemble | 73.24/2.774 | 73.92/2.675 | 70.29/2.939 |
| | MC Dropout | 69.16/3.082 | 69.79/3.041 | 64.05/3.449 |

Table 7: ImageNet results for ResNet18 with the accuracy and ECE metric (accuracy/ECE). The pruning sparsities refer to the number of pruned layers, and the pruned models are fine-tuned with 150 epochs. The ECE metric is scaled by $1e^{-1}$.

| Pruning Method | Uncertainty Method | Sparsities | | |
|---|---|---|---|---|
| | | 0.0 | 0.2 | 0.6 |
| Random Pruning | Softmax | 69.76/0.142 | 66.27/0.168 | 55.77/0.250 |
| | Ensemble | 73.24/2.099 | 69.43/2.414 | 60.17/3.119 |
| | MC Dropout | 69.05/2.342 | 62.72/2.872 | 27.40/6.462 |
| L1 Pruning | Softmax | 69.76/0.142 | 67.12/0.125 | 56.00/0.249 |
| | Ensemble | 73.24/2.099 | 70.24/2.303 | 60.54/3.122 |
| | MC Dropout | 69.05/2.342 | 64.82/2.687 | 21.28/7.152 |
| Batch Norm Pruning | Softmax | 69.76/0.142 | 65.98/0.152 | 55.93/0.272 |
| | Ensemble | 73.24/2.099 | 69.28/2.443 | 58.48/3.701 |
| | MC Dropout | 69.05/2.342 | 62.57/2.852 | 11.64/8.187 |
| HRank Pruning | Softmax | 69.76/0.142 | 66.63/0.136 | 55.99/0.274 |
| | Ensemble | 73.24/2.099 | 70.12/2.288 | 60.47/3.225 |
| | MC Dropout | 69.05/2.342 | 63.18/2.974 | 27.79/6.511 |
| ResRep Pruning | Softmax | 69.76/0.142 | 69.88/0.405 | 65.82/0.205 |
| | Ensemble | 73.24/2.099 | 73.92/2.151 | 70.29/2.382 |
| | MC Dropout | 69.05/2.342 | 69.84/2.332 | 63.95/3.009 |

