# OpenReview forum: "Effects of Neural Network Parameter Pruning on Uncertainty Estimation"
_TMLR — Withdrawn by Authors_

### Review · Reviewer_6QWG · 2024-04-04

**Summary Of Contributions:**

This work proposes to study how parameter pruning affects uncertainty quantification for image classification models. It considers various parameter pruning methods, including random pruning, L1 pruning, batch normalization pruning, HRank pruning, ResPep pruning; and three uncertainty quantification techniques, including maximum softmax probability, bootstrapped ensembles, and Monte Carlo dropout.

This work uses two metrics (AUSE and ECE) to find that uncertainty estimation performance degrades as more parameters are pruned.

**Audience:**

No

**Broader Impact Concerns:**

Discussions on limitations are recommended. To my knowledge, there are no ethical concerns.

**Claims And Evidence:**

Yes

**Requested Changes:**

- Please experiment with large models, e.g., Resnet152 and Vision transformer models.
- Please consider other more commonly used uncertainty evaluation metrics or justify why AUSE is the best choice.
- Please more formally define notations in section 5 - proposed uncertainty metric.
- Please consider re-calibrating the model before evaluating its uncertainty performance or justify why including re-calibration is unnecessary or unreasonable.
- Please provide some insights in the experiment section, especially for interesting results like Fig.3(d).
- In my opinion, results on Imagenet are more interesting. Please consider moving the Imagenet results to the manuscript.
- Please consider reporting the average inference time, at least in the appendix.

**Strengths And Weaknesses:**

## Strength
- This work studies a novel problem, connecting parameter pruning to uncertainty quantification.
- This work considers multiple pruning and uncertainty quantification techniques.
- This work proposes to utilize AUSE to measure uncertainty quantification in image classification problems.
## Weakness
- The experiment uses Resnet18 and VGG19, which have relative small sizes. The motivation for pruning these models is weak, and findings are less interesting to the community.
- While this work proposes to use AUSE, it is unclear why more commonly used metrics, including AUROC, AUPRC, and AUARC, are not preferred.
- When uncertainty is considered in model utilization, it is desired to re-calibrate the model via techniques like temperature scaling or histogram scaling. Measuring uncalibrated models is less convincing.
- In the experiment section, limited insights are provided. Abnormal behaviors, like the one demonstrated in Fig.3(d) is not fully analyzed.
- When describing AUSE, it is difficult to clearly understand notations like Oracle, Actual, uncertainty of the predicted class, the true uncertainty, etc.
- Although not fully studied, the finding that a pruned model would induce suboptimal uncertainty performance is not surprising.
## Some questions:
- In the experiments, are models fine-tuned after pruning?

---

### Review · Reviewer_zye7 · 2024-04-09

**Summary Of Contributions:**

The authors empirically investigate the effect of five channel-pruning methods on three uncertainty measures. The pruning methods use various strategies to identify channels of a convolutional layer that provide little information and can hence, be pruned. However, four of the five evaluated pruning methods do not update the pruning selection simultaneous to the network weights, which makes additional training necessary after the pruning. As uncertainty measures, the authors use max-softmax confidence, MC dropout, and ensembles. The results are measured via the Area Under the Sparsification Error (AUSE) metric (that has so far been used for uncertainty measures in optimal flow estimation) and the Expected Calibration Error (ECE).
The experiments are conducted on CIFAR-100 and ImageNet. The general tendency, that is visible from the results, is that the accuracy drops and the AUSE increases with increasing pruning rates. Likewise, the ECE typically increases with increasing pruning rates.

**Audience:**

Yes

**Claims And Evidence:**

No

**Requested Changes:**

Critical for acceptance would have been (I believe it's too much to incorporate in a revision):
* Identify a link between the effect of sparsity in networks and their uncertainty evaluation. Focus on evaluating this link.
* Put the chosen uncertainty measures in context - on what assumptions do they work, when do they fail, what alternatives are there? In particular, I wonder why not more established uncertainty measures are used, such as the energy score.
* Put the chosen pruning methods in context - when are they applicable, because the ones presented drop the ACC by a lot, is there nothing better out there?
* Either really explain and motivate the AUSE, or do some more established experiments, for example with OOD data. The plots are also really difficult to read. I can't derive from the marker size which pruning factor has been applied. Why not make a table where each row reflects the pruning scale, ACC, uncertainty evaluation and other metrics?
* Discuss and evaluate other explanations of your results. In addition to the potential ACC - uncertainty relation, could the results not possibly also be caused by the smaller architecture used for the pruned networks? How can we actually conclude that the pruning causes the observed effect and not other circumstances?

**Strengths And Weaknesses:**

Strengths:
* interesting premise of potential results
* Paper is good to follow in terms of reading

Weaknesses:
* the experiments are not designed to reject or provide evidence for a specific hypothesis
* the interpretation of the results is unclear - the observed effects are not linked to any specific properties of the pruning mechanism or the uncertainty measurement and could also just reflect correlations between, e.g., the accuracy and the uncertainty measure
* the chosen evaluation metrics are not well motivated. In particular, the definition of the "true uncertainty" as 1- softmax confidence of the actual class has flaws, inherited by the flaws of the softmax confidence.
* metrics and uncertainty estimation methods are not critically reflected. There are well-known major issues with softmax confidence-based [1] metrics, and the ECE [2].

Evaluating the effect of pruning/sparsification of neural networks on their uncertainty representation is generally very interesting. However, I am missing in this work a definable link between the properties that the uncertainty measure uses, and the effect that pruning has on the network. As it is now, the paper reads as _a_ comparison between a bunch of uncertainty measures and channel-pruning methods. I can't derive a proper conclusion from the experiments, and everything I get from the author's conclusion is that the uncertainty estimation performance drops with the accuracy. Hence, the question arises whether the observed effect is not merely a side-effect of an uncertainty estimation that deteriorates with the accuracy.
The main evaluation metric AUSE is hard to understand in the presented form, I am missing a formula for its computation in the main text. The authors also do not motivate the need for this metric. Typically, the uncertainty is measured using Out-of-distribution OOD datasets, where the True Positive Rate of detected OOD samples is then computed for a fixed False Positive Rate of 5%. Or the AUROC is computed. This procedure has the advantage that the uncertainty metric is evaluated in terms of its usability to actually reflect what kind of images the network hasn't seen. In the AUSE metric (according to the pseudocode in the appendix), the uncertainty of the prediction is evaluated against the uncertainty of the "true" label. Isn't that clear then that the AUSE is hence highly dependent on the accuracy?

[1] Gawlikowski, J., Tassi, C.R.N., Ali, M. et al. A survey of uncertainty in deep neural networks. Artif Intell Rev 56 (Suppl 1), 1513–1589 (2023). https://doi.org/10.1007/s10462-023-10562-9

[2] Liu, Jeremiah, et al. "Simple and principled uncertainty estimation with deterministic deep learning via distance awareness." Advances in neural information processing systems 33 (2020): 7498-7512.

---

### Review · Reviewer_6tte · 2024-04-14

**Summary Of Contributions:**

The authors present a study of the influence on the uncertainty estimation when pruning the parameters of image classification network. They study on 5 pruning methods and 3 uncertainty methods. The authors also propose the uncertainty metric, named AUSE, and use it for studying the influences. They show that the pruning generally affects the uncertainty estimation in a negative manner.

**Audience:**

No

**Claims And Evidence:**

No

**Requested Changes:**

The proposals are mainly described in the Weaknesses. I request the authors to change (or add) the explanations/experiments/plots mentioned above.

minor:
- Some of the word "excepted" should be changed to "expected" when describing ECE (e.g. in Abstract and in Introduction).

**Strengths And Weaknesses:**

Strengths:
- The paper addresses an important task of understanding how uncertainty estimation is affected by pruning the model. This study would be more important when employing large models.

Weaknesses:
- The experiments could be more comprehensive. Including additional uncertainty estimation methods such as temperature scaling [A], last-layer-only Bayesian inference [B], or LaplaceApprox [C] would enhance the representation of uncertainty methods.

- The rationale behind introducing the AUSE metric needs further clarification. It would be beneficial to explain the purpose of this metric, compare it with previously proposed metrics (e.g., proper scoring rules or ECE), and elaborate on its advantages.

- Regarding the metric, authors can also try the experiments on metrics popularly used in the literature, like proper scoring rules or on out-of-distribution data [D]. Is there a reason for deciding not to use these metrics?

- The authors normalized the uncertainty of MC dropout and bootstrapped ensembles into the range [0, 1] in order to have ECE. However, does it make sense to apply normalization on those uncertainy values? The logic behind ECE is that the uncertainty should be a predictive probability of the target class, which I don't think it is when uncertainty by MC dropout is normalized.

- What could be the reason behind the experimental observations, e.g. why ResRep pruning shows a different behavior?

- The Figure 2-4 could be improved. Currently we can just assume that the highest accuracy point in each line corresponds to the non-pruned version.

- It would be better broaden the scope of experiments to provide a more comprehensive understanding of the relationship between uncertainty estimation and pruning.


[A] On calibration of modern neural networks, ICML 2017.

[B] Deep Bayesian Bandits Show: An Empirical Comparison of Bayesian Deep Networks for Thompson Sampling, ICLR 2018.

[C] A Scalable Laplace Approximation for Neural Networks, ICLR 2018.

[D] Can You Trust Your Model's Uncertainty? Evaluating Predictive Uncertainty Under Dataset Shift, NeurIPS 2019.

---

### Review · Reviewer_2gTW · 2024-04-14

**Summary Of Contributions:**

This work investigates the impact of parameter pruning on uncertainty estimation in deep neural networks, specifically in image recognition tasks. The work proposes to analyze the effect of pruning of models not only on accuracy performance but also on uncertainty estimation using different combinations of network architectures, pruning algorithms, and uncertainty estimation methods. This paper is an empirical study of the effect of model compressions on the reliability of the produced models.

**Audience:**

Yes

**Claims And Evidence:**

No

**Requested Changes:**

Please see weaknesses part of the review.

**Strengths And Weaknesses:**

Strengths:

The main purpose of this paper is to conduct an empirical analysis of the effect of model pruning on the uncertainty of the compressed models, not only the accuracy performance. In this setting, the paper showcases a decent set of experiments and metrics.

Weaknesses:

This paper has a severe novelty limitation. The paper claims that "no study has yet examined [this] issue in the context of parameter pruning", but concludes that the effect of pruning/compression on model performance has a high correlation between the effect on the accuracy and on the uncertainty. To this end, most findings were trivial. Maybe the only thing that was worth noticing is that ResRep Pruning is more suitable for pruning in general, which is yet not new [R1].

This paper lacks a thorough analysis of the effect of model compression on the final performance with detailed discussion. As the empirical investigation of this work has shown a great correlation between uncertainty and accuracy performance, the investigation should refer to the work in [R2] and highlight what was novel to find. Similarly, why studying pruning has a different interest than studying quantization [R3]. The sole focus of this work on pruning is unjustified.

The second contribution of having "Area Under the Sparsification Error (AUSE) metric as an uncertainty measure for image classification to quantify uncertainty prediction" is an over-statement. The difference between this work's AUSE implementation and the one in [R4] needs to be clarified.

The third contribution," that model pruning affects the model performance along with the uncertainty estimation," is not new. Additionally, it has been shown previously that pruning to a certain low sparsity value can actually preserve the original performance while lowering the complexity of the model. This is equivalent to reducing the variance of deep models, which explains the better calibration performance of the compressed models. Otherwise, pruning/compression techniques would not have been adopted in the Deep Learning field. This comes back to the original statement that this work lacks novelty as all observations were trivial.

This work explains that it investigates different experimental settings for generalization purposes. The use of more than one architecture or dataset is justified for diversity purposes; however, investigating five pruning techniques and three uncertainty estimation methods. All the pruning techniques show similar conclusions. There are no additional insights on the investigated methods beyond their original work. Second, it is unjustified to include "accuracy" as an uncertainty measure in this work as it has been extensively evaluated in the state of the art.

[R1]: Ding, Xiaohan, et al. "Resrep: Lossless cnn pruning via decoupling remembering and forgetting." Proceedings of the IEEE/CVF international conference on computer vision. 2021.
[R2]: Hooker, Sara, et al. "What do compressed deep neural networks forget?." arXiv preprint arXiv:1911.05248 (2019).
[R3]: Ferianc, Martin, et al. "On the effects of quantisation on model uncertainty in bayesian neural networks." Uncertainty in Artificial Intelligence. PMLR, 2021.
[R4]: Ilg, Eddy, et al. "Uncertainty estimates and multi-hypotheses networks for optical flow." Proceedings of the European Conference on Computer Vision (ECCV). 2018.

---

### Author Response · Authors · 2024-04-27
**Thank you**

Dear Reviewers and Action Editor,

We would like to express our thanks to the Action Editor and the reviewers for their valuable time and effort in providing insightful and helpful suggestions. Given the reviews, we decided to further work on the manuscript before going through another round of reviewing.

Once again, we thank the Action Editor and the reviewers for their time and effort in reading and analyzing this manuscript.

Sincerely,

the authors

---

### Note · Authors · 2024-04-27

I have read and agree with the venue's withdrawal policy on behalf of myself and my co-authors.